

# Sex differences in association football: a scoping review

Wangyang Xu[1], Diyan Zhang[2] and Xinbi Zhang[3,4]

[1] School of Athletic Performance, Shanghai University of Sport, Shanghai, China
[2] Physical Education Institute, Henan Normal University, Xinxiang, China
[3] The Center of Neuroscience and Sports, Capital University of Physical Education and Sports, Beijing, China
[4] School of Kinesiology and Health, Capital University of Physical Education and Sports, Beijing, China

Corresponding author
Wangyang Xu,
18161993428@163.com

## ABSTRACT

**Background:** Despite some reviews examining sex differences in football within specific themes, a comprehensive, integrated overview of sex differences in football is lacking. This scoping review aimed to: (1) synthesize existing evidence regarding sex differences in elite football players; (2) identify research gaps to provide direction for future studies.

**Methodology:** The protocol adhered to the PRISMA Extension for Scoping Reviews guidelines. The searches were conducted on October 17, 2024, in Scopus, PubMed, ScienceDirect, and Web of Science (Core Collection). The risk of bias was assessed using the Revised Risk of Bias Assessment Tool for Nonrandomized Studies of Interventions (RoBANS 2). A narrative synthesis was performed to summarize the main findings.

**Results:** A total of 80 studies met the eligibility criteria and were included in the review, encompassing 4,896 players (2,226 female, 2,670 male) and 234 matches (99 female, 135 male). Seventy-two studies (90%) did not report female participants' menstrual cycles or contraceptive medication use. Only six studies exclusively used "sex"-related terminology, with not a single study using "gender"-related terminology alone. Seventy-four studies (93%) presented various degrees of mixed usage of the terms "sex" and "gender". All included studies were categorized into eight themes according to their research focus: Anthropometrics, Muscle and Joint Movements, Physiological Response, Physical Performance, Technical Performance, Match Performance, Psychological and Behavioral Performance, and Nutrition/Recovery/Sleep.

**Conclusions:** Perhaps due to a combination of innate biological factors and acquired dietary habits, female players, who typically have a higher body fat percentage and lower lean body mass, exhibit performance gaps compared to male players in terms of speed, strength, and endurance. These differences further affect their technical skills and match performance. However, while these absolute differences exist, they often diminish or even disappear when data are standardized against certain anthropometric or physiological metrics. This underscores the importance of developing individualized analytical methods and evaluation criteria tailored to female players. Future studies should carefully define sex-specific inclusion/exclusion

criteria and select appropriate sex and gender terminology to minimize bias and enhance study quality.

# INTRODUCTION

Despite the fact that female football has not yet achieved the same level of influence as male football, the significant increase in participation and widespread recognition by international governing bodies in recent years have drawn growing attention from global sports scholars (*Pfister, 2015*). However, compared to the relatively mature research framework in male football, studies on female football and its related aspects are still in the developmental stage (*Kryger et al., 2022*). It remains uncertain and often questioned whether the knowledge accumulated from male football can be effectively applied to female football to gain a comprehensive and accurate understanding of the latter (*Bradley & Vescovi, 2015*; *Kryger et al., 2022*). Concurrently, as research in female football expands (*Armendáriz, Spyrou & Alcaraz, 2024*; *Barreira, da Silva Junior & de Souza, 2024*), numerous studies have focused on the sex differences among football players (*Cardoso de Araujo et al., 2018*; *McFadden et al., 2024*; *Mujika et al., 2009*). These sex difference studies facilitate a deeper understanding of the unique characteristics and developmental needs of female football from a comparative perspective. By studying the differences between male and female football players across various aspects, these studies help to uncover the distinct challenges and potential advantages faced by female football in training and competition. The findings not only provide scientific basis for the training and matches of female football but also offer valuable references for its promotion and development. Furthermore, these studies contribute to eliminating sex bias, promoting sex equality, and driving the healthy development of female football.

Generally, sex differences may stem from biological differences or differences in physical activity (*Miller et al., 1993*). For instance, while significant performance gaps between sexes exist in the general population, these diminish when controlling for various attributes (*Lewis, Kamon & Hodgson, 1986*). A study by *Freedson et al. (1979)* found that when male and female participants were matched for VO2 max, sex differences in cardiac output and related metrics vanished after adjusting for lean body mass. Additionally, training has been shown to alter neuromuscular control (*Barendrecht et al., 2011*; *Mizner, Kawaguchi & Chmielewski, 2008*; *Noyes et al., 2005*). Skilled players exhibit similar movement patterns regardless of sex, due to the homogenizing effect of training (*Bruton, O'Dwyer & Adams, 2013*). Given the theoretical perspectives and evidence, and the focus on sex differences in elite football players (*McKay et al., 2022*), it is reasonable to assume that both sexes undergo tailored physical stimulation and adaptation processes during training and competition. These processes are designed according to sex characteristics and produce sex-specific responses. Through a thorough investigation of sex differences in

elite football players, we can better understand the characteristics and advantages of male and female players in football, which will inform more targeted guidance and support.

Previous studies have reviewed sex differences in football under specific research themes (*Boyne et al., 2021*; *Dave et al., 2022*; *Fältström, Hägglund & Kvist, 2024*; *Gulbrandsen et al., 2019*; *Robles-Palazón et al., 2021*; *Xiao et al., 2022*). For example, *Robles-Palazón et al. (2021)* reviewed studies on injuries in male and female youth football players. They found that males tend to sustain predominantly muscle injuries to the thigh, while females sustain predominantly joint and ligament injuries to the knee and ankle, reinforcing the need for different targeted management strategies in male and female youth players. *Boyne et al. (2021)* summarized sex-based differences in kicking biomechanics in soccer, proposing that skill level within each sex may play a more important role in kicking performance than differences between the sexes. However, there is a notable absence of a review that provides a thorough and integrated overview of the existing literature on sex differences among football players.

Moreover, in investigating sex differences, accurately distinguishing between 'sex' and 'gender' is a fundamental prerequisite in scientific research design. Imprecision in terminology may lead to misunderstandings of results and subsequently affect the translation of research into practical applications (*Moores et al., 2023*). In terms of definition, a study that selects participants based on their biological characteristics will be considered appropriate to use the term sex, with the primary binary classification being male and female. In this context, researchers are examining sex differences rather than gender differences. In contrast, if judgments are made about non-biological characteristics or social categories, gender will be used as the reference, with the primary binary classification being man and woman (*Deaux, 1985*; *Heidari et al., 2016*). For the purpose of this study, the focus will be on sex rather than gender. This study is intended for sports scientists, performance analysts, football coaches, athletic trainers, and sports medicine practitioners who seek to understand sex-based differences in football performance. The synthesized evidence provides information for future methodological approaches and research priorities in female-specific performance analysis within football.

Therefore, this study is a scoping review of sex differences in elite football players, aiming to: (1) synthesize existing evidence regarding sex differences in elite football players; (2) identify research gaps to provide direction for future studies.

## METHODS

### Study design and protocol registration

This scoping review followed the PRISMA Extension for Scoping Reviews (PRISMA-ScR) guidelines (*Tricco et al., 2018*). The protocol was prospectively registered on the Open Science Framework (https://osf.io/gnmky) on 30 October 2024.

### Eligibility criteria

The inclusion criteria were guided by the Participants, Intervention/Exposure, Comparators, Outcomes, and Study Design (PICOS/PECOS) framework (*Sarmento et al., 2024*) as follows: (1) Participants—healthy adult male and female football players,

categorized from Tier 3 to Tier 5 based on the Participation Classification Framework (*McKay et al., 2022*), including Highly Trained/National Level (Tier 3), Elite/International Level (Tier 4), and World Class (Tier 5); (2) Intervention/Exposure—any measure, intervention, or exposure relevant to sports science or sports medicine; (3) Comparators— at least two groups of football players divided by sex; (4) Outcomes—physical measures, physiological measures, technical assessments, and psychological assessments; (5) Study Design—no restrictions on the types of study designs eligible for inclusion.

Studies were excluded if they: (1) did not include original data; (2) were not available in English and full text.

## Information sources and search strategy

This review only included peer-reviewed original research studies in the English language. The following databases were searched: Scopus, PubMed, ScienceDirect, and Web of Science (Core Collection). These searches encompassed relevant publications available up to 17 October 2024. In addition, manual searches were conducted on the reference lists of the included studies to identify potentially relevant studies. The search strategy is detailed in Table S1.

## Selection process

Search results were exported to EndNote X9, where one researcher (WX) removed all duplicates. Two researchers (WX, XZ) independently conducted the screening process. Titles and abstracts were first subjected to a preliminary review, discarding studies that were unrelated to the review's subject matter. Subsequently, the full texts of the remaining studies were read, and those that failed to meet the inclusion criteria were excluded. In case of discrepancies, a third researcher (DZ) was consulted to achieve consensus.

## Data extraction

For each study, key information from the included texts was extracted into a custom-designed data extraction form. This form was scrutinized by the research team and pretested by all researchers before implementation to guarantee the accurate collection of information. Two researchers (WX and XZ) independently extracted data using the form, while a third researcher (DZ) cross-checked the information to ensure its accuracy and reliability.

## Data items

In light of the heterogeneity among the included studies, a comprehensive set of data was collected for each one. The data were extracted in the following order: the first author's name and country, publication year, study design, experimental protocol, samples, main results, and usage of sex/gender terms.

This review tallies the occurrences of the terms "sex(es)", "gender(s)", "male(s)", "female(s)", "man(men)", and "woman(women)" within each study text included to ascertain the current usage of gender terminology in discussions about sex differences among elite football players.

### Risk of bias

The risk of bias was assessed independently by two researchers (WX and XZ), with disagreements resolved through reanalysis. If consensus was unattainable, a third researcher (DZ) rendered the final decision. The Revised Risk of Bias Assessment Tool for Nonrandomized Studies of Interventions (RoBANS 2) was utilized to assess the risk of bias in the studies that were included in this review (*Ma et al., 2020*; *Seo et al., 2023*). It has been assessed to have acceptable feasibility, fair-to-moderate reliability, and adequate construct validity (*Seo et al., 2023*). It encompasses eight domains, namely: comparability of the target group, target group selection, confounders, measurement of intervention/exposure, blinding of assessors, outcome assessment, incomplete outcome data, selective outcome reporting. Each of these domains is categorized into "low", "high", or "unclear" risk of bias.

### Data synthesis

Due to substantial heterogeneity among the included studies, a narrative synthesis was adopted to facilitate a nuanced interpretation of findings across diverse study contexts. This approach involved categorizing the studies according to their methodologies, key results, and primary discussion themes. To ensure classification accuracy and objectivity, two researchers (WX and XZ) independently performed the categorization. A third researcher (DZ) subsequently cross-validated the groupings. Any discrepancies in the initial classifications were discussed and resolved by consensus among all three researchers.

## RESULTS

### Search results

Figure 1 illustrates the process of searching and selecting studies. Initially, 3,027 studies were retrieved from the database. After removing 1,604 duplicates, 1,423 studies underwent title and abstract screening, excluding 1,270 records. Of the remaining 153 studies, six were excluded due to unretrievable full texts. Following full-text analysis, 73 studies were excluded. Six studies were identified through reference list screening, resulting in a total of 80 studies included in the scoping review.

### Description and characteristics of the included studies

All included studies are cross-sectional, encompassing a total of 2,226 female football players (average age: 18–26 years) and 2,670 male football players (18–28 years), as well as 99 female and 135 male matches. Table S2 provides a summary of the included studies. Figure 2A shows the distribution of studies by publication year. Of the 80 studies, the earliest one was published in 1998, and there were no publications on sex differences among healthy adult elite football players in the years 1999, 2001, 2004, and 2005. Notably, 35% ($n = 28$) of the studies were published in the past three years (from January 1, 2022 to October 17, 2024). Figure 2B presents the distribution of the first authors' countries of origin. Fifty-one percent ($n = 41$) of the studies had first authors affiliated with institutions in Europe. The most frequent countries of affiliation for the first authors were the USA and
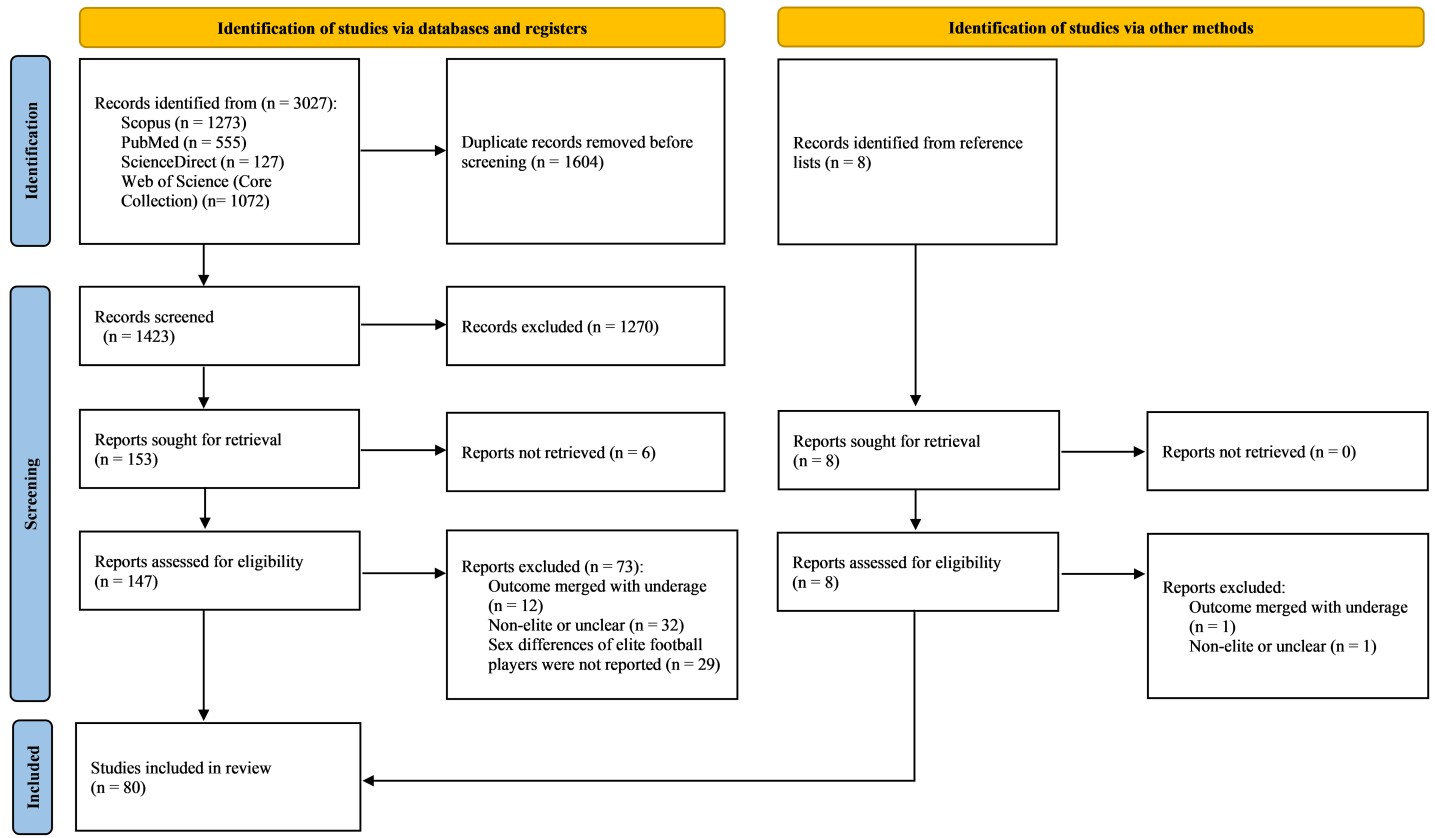

**Figure 1  Flow diagram of the search and selection process (*n* = 80).**

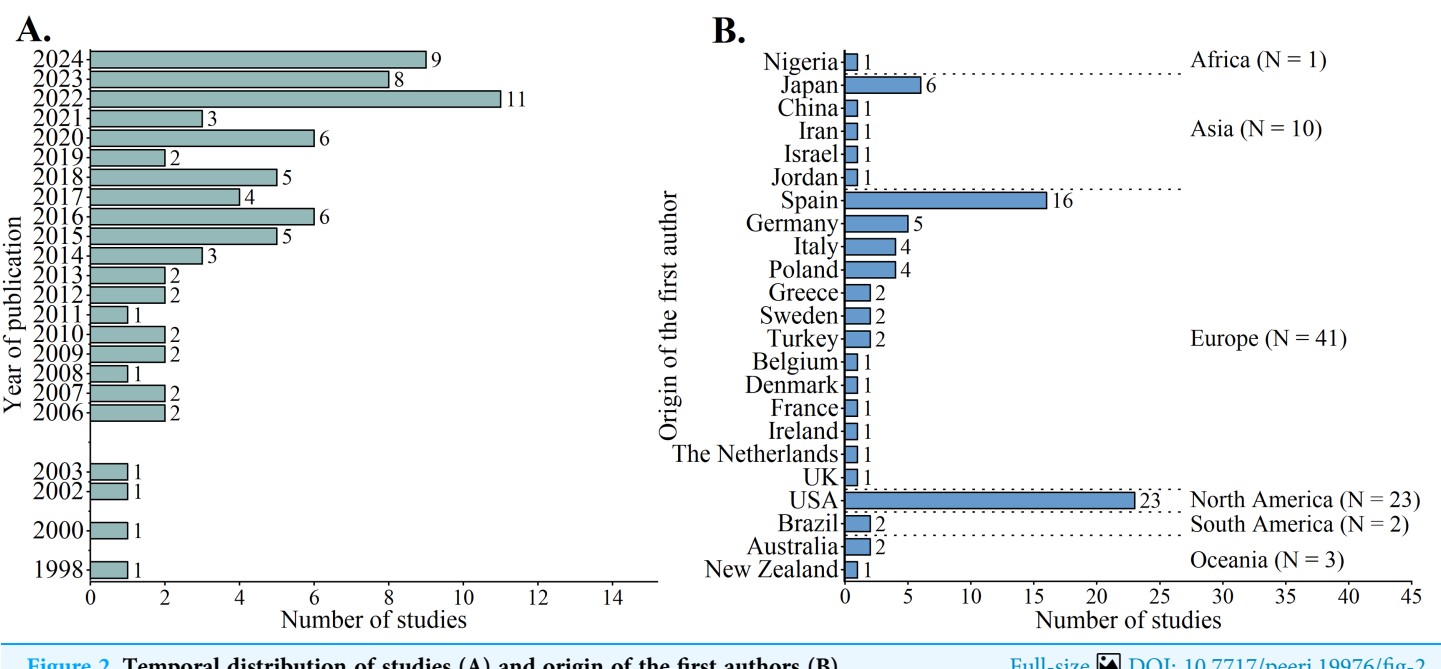

**Figure 2  Temporal distribution of studies (A) and origin of the first authors (B).**

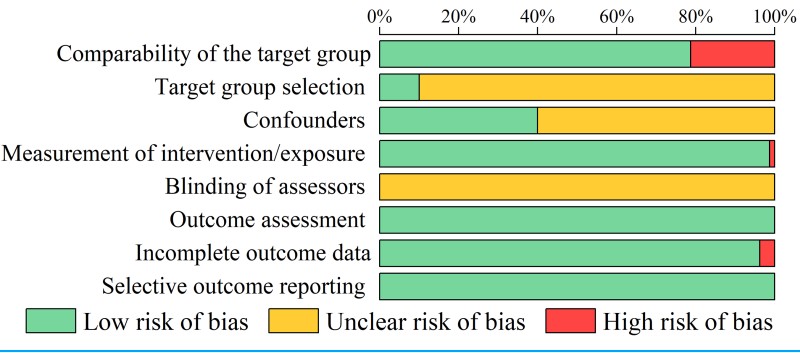

**Figure 3** The risk of bias percentage distribution in the included studies.

Spain, accounting for 29% ($n$ = 23) and 20% ($n$ = 16) of the total number of studies, respectively.

## Risk of bias assessment

Table S3 presents the full results of the risk of bias assessment for the included studies based on the RoBANS 2. The risk of bias percentage distribution for the included studies is illustrated in Fig. 3. All studies presented a low risk of bias in the outcome assessment and selective outcome reporting, but the risk of bias in the blinding of assessors was unclear, as none of the studies reported blinding. Seventeen studies (21%) presented a high risk of bias in the comparability of the target group, as the male and female participants were from different levels of leagues. Seventy-two studies (90%) presented an unclear risk of bias in the selection of the target group due to the lack of reporting on female participants' menstrual cycles and the use of contraceptive medications. Forty-eight studies (60%) presented an unclear risk of bias in the confounders, as there was no information provided on participants' sleep, exercise, and eating habits prior to testing. One study presented a high risk of bias in the measurement of intervention/exposure because urine samples from male and female participants were collected during different months of the year (*Rodas et al., 2022*). Three studies presented a high risk of bias in incomplete outcome data. Two of these studies did not collect data from all participants at each test item or time point (*Cardoso de Araujo et al., 2018*; *Rodas et al., 2022*), and one reported the loss of some data due to a computer malfunction (*Putukian, Echemendia & Mackin, 2000*).

## The usage of sex and gender terminology

Table S2 and Fig. 4 illustrate the usage of gender terminology in the included studies. Only six studies exclusively used "sex"-related terminology, with not a single study using "gender"-related terminology alone (*Dent et al., 2015*; *Dolci et al., 2021*; *Iitake et al., 2022*; *Langdon et al., 2022*; *Navandar, Kipp & Navarro, 2022*; *Siegle & Lames, 2012*). The remaining 74 studies presented various degrees of mixed usage of the terms "sex" and "gender", constituting 93% of the total number of the included studies. The most prevalent category comprised studies that used all "sex" and "gender" terms simultaneously, as well

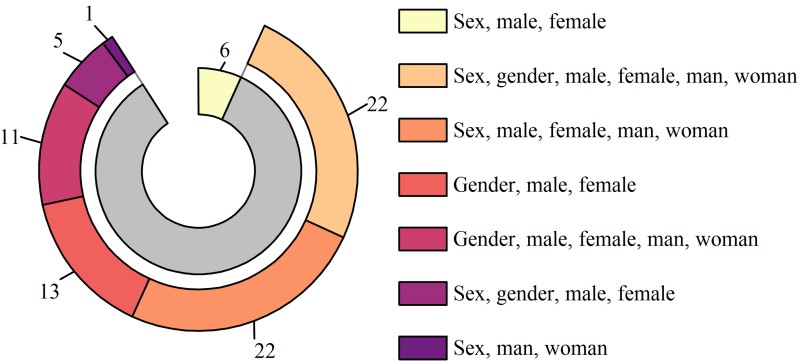

**Figure 4** The distribution of sex and gender terminology.

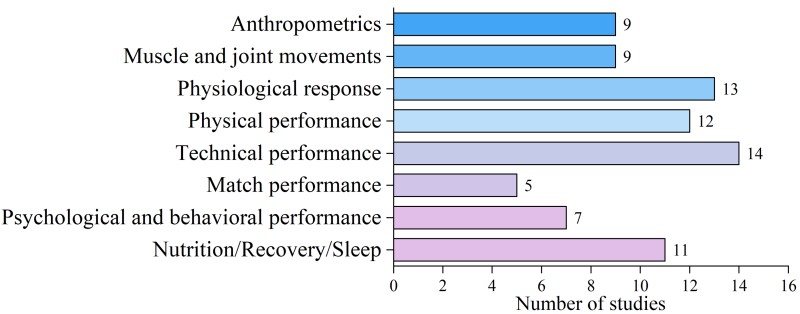

**Figure 5** The classification of the research themes.

as those that used all terms except for the word "gender(s)", collectively accounting for 55% of all studies, with 22 studies in each group.

## Research themes

Based on the methods and main results of each study, as well as the main focus of the discussion sections in the articles, the researchers (WX, XZ, and DZ) reached a consensus to categorize the 80 included studies into eight different research themes, as illustrated in Fig. 5.

*Anthropometrics* focuses on sex differences in body shape, body composition, and skeletal characteristics.

*Muscle and Joint Movements* discusses sex differences in muscle activity and joint movements under biomechanical assessment.

*Physiological Response* focuses on the internal body responses of football players of different sexes to various exercise stimuli.

*Physical Performance* discusses sex differences in performance on football-specific tests such as jumping, sprinting, agility, and intermittent endurance.

*Technical Performance* focuses on sex differences in kicking techniques, particularly the instep kick, and heading ability.

*Match Performance* discusses sex differences in technical events, running performance, and match interruptions.

*Psychological and Behavioral Performance* focuses on sex differences in mental health, motivation, visual attention, and behavioral patterns.

*Nutrition/Recovery/Sleep* discusses sex differences in nutrient intake, recovery from fatigue, and sleep quality and sleepiness.

### Anthropometrics (*n* = 9)

*Petri et al. (2024a)* suggested that female players are categorized as balanced mesomorphs, whereas male players are classified as ectomorphic mesomorphs. Besides height, weight, and body mass index, where female players typically do not have an advantage, they exhibit a higher percentage of body fat (%BF) and a lower fat-free mass than male players (*Ahmad & Abu Al Haija, 2024*; *Mascherini et al., 2017*; *Petri et al., 2024b*; *Tornero-Aguilera, Villegas-Mora & Clemente-Suarez, 2022*; *Toro-Román et al., 2023a*). *Schons et al. (2022)* found that total skinfold thickness negatively correlated with drop jump, Yo-Yo IR1 test performance, and sprint times in male players, with the same trend observed in female players for the first two measures.

Moreover, two studies have investigated sex differences in skeletal characteristics among elite football players (*Baker et al., 2020*; *Hedt et al., 2022*). *Hedt et al. (2022)* observed that female players had lower values for pelvic width, femur length, and tibia length compared to males. *Baker et al. (2020)* found that, compared to female players, male players have greater whole-body and hip areal bone mineral density, as well as larger bone size, strength, and volumetric bone mineral density at the tibial site. However, these sex differences were mitigated when the influence of bone free lean body mass and fat mass was accounted for in regression analyses. Additionally, the tibia of the dominant leg in female players showed a greater capacity for bone adaptation, whereas the non-dominant leg tibia of male players demonstrated a higher adaptive capability.

### Muscle and joint movement (*n* = 9)

Given that football is a lower-limb-dominant sport, research on upper-limb strength differences between male and female players is relatively scarce. *Ichinose et al. (1998)* found that female players had lower elbow extension torque than males, but this discrepancy disappeared when adjusted for the cross-sectional area of triceps brachii. In the lower limbs, both sexes of players exhibited limited hip rotation, with females being more prone to internal rotation. Female players had weaker abdominal and non-dominant hip abduction strength (*Brophy et al., 2009*). During single-leg landings, female players had lower gluteus medius activity than males (*Hart et al., 2007*), which is crucial for pelvic stability and resistance to hip adduction/internal rotation (*Anderson & Pandy, 2003*). Considering the dynamic coupling between segments of the kinetic chain, asymmetry in

hip abductor and adductor muscle activation may contribute to the valgus knee position in female players' landing patterns (*Hewett et al., 2006*). Moreover, *Makaraci et al. (2024)* found that functional movement screen scores were positively correlated with postural stability in both sexes.

Ultrasonography revealed that female players have thinner muscle thickness and a smaller fascicle angle in the vastus lateralis and medial gastrocnemius than males, with no fascicle length differences (*Kanehisa et al., 2003*). Female players exhibit lower peak torques for both quadriceps and hamstrings than males, even when normalized by body weight (*Burfeind, Hong & Stavrianeas, 2012*; *Yilmaz et al., 2023*; *Zebis et al., 2011*). A lower hamstring-to-quadriceps (H/Q) peak torque ratio is a risk factor for injury (*Andrade et al., 2012*). The greater quadriceps strength relative to hamstrings after menarche in female players may increase their risk of anterior cruciate ligament (ACL) injury (*Ahmad et al., 2006*). However, studies have not reported sex differences in H/Q ratios or the time to achieve peak torque, with these ratios remaining within a healthy range (*Burfeind, Hong & Stavrianeas, 2012*; *Yilmaz et al., 2023*; *Zebis et al., 2011*). This may be attributed to the high level of play and intensive, long-term training among elite female players, which mitigates these disparities. Nevertheless, *Yilmaz et al. (2023)* found that female players had significantly greater bilateral asymmetry in hamstring strength at $60°/s$ and $180°/s$. In summary, poor core control, hip muscle imbalance, and bilateral asymmetry may increase the risk of ACL injuries and other injuries in female players.

## Physiological response (*n* = 13)

A long period of intense football training increases left ventricle volume and wall thickness, thereby enhancing cardiac function, including an increase in stroke volume and cardiac output during exercise (*Sansonio de Morais et al., 2018*; *Steding et al., 2010*). This boosts peripheral blood flow and oxygen delivery to metabolically active muscles (*Abergel et al., 2004*; *D'Ascenzi et al., 2014*). Female players have smaller left ventricle dimensions and lower parameters compared to male players (*Mascherini, Petri & Galanti, 2018*; *Sansonio de Morais et al., 2018*; *Steding et al., 2010*). These differences may be due to factors such as smaller body surface area, lower peak systolic blood pressure during exercise, and lower levels of androgenic hormones in females (*Di Paolo & Pelliccia, 2007*). Even after adjusting for body surface area, the total heart volume of female players remains smaller than that of male players (*Steding et al., 2010*). However, the sex differences in absolute heart diameter, left ventricular septal, and posterior wall thickness are no longer apparent (*Sansonio de Morais et al., 2018*).

Introduced in 1974 (*Ayalon, Inbar & Bar-Or, 1974*), the Wingate Anaerobic Test (WAnT) is a widely studied anaerobic assessment (*Bar-Or, 1987*) and is now incorporated into high-intensity interval training (HIIT) protocols (*Burgomaster et al., 2005*; *Freese, Gist & Cureton, 2013*; *Williams & Kraemer, 2015*). *Magal et al. (2020)* conducted a study involving six 30-s WAnTs, each followed by a 4-min recovery period. After using the first WAnT as the baseline and adjusting for fat-free mass, the authors calculated the percentage change in metrics for WAnTs 2–6 and found no sex differences in peak or mean power. For aerobic exercise, three studies investigated the physiological responses of

football players to a 60-min outdoor run (*Chamera et al., 2014, 2015; Kostrzewa-Nowak et al., 2015*). Pre-exercise markers of muscle status were similar between sexes, but post-exercise changes differed. The levels of aspartate aminotransferase and alanine aminotransferase increased in the blood of female players following exercise, a change not observed in male players (*Chamera et al., 2014*). Creatine kinase (CK)-MB activity increased in both sexes, while total lactate dehydrogenase activity increased only in female players, and α-hydroxybutyrate dehydrogenase activity only in male players (*Chamera et al., 2015*). Both sexes experienced increased total blood protein levels, with a significant rise in C-reactive protein (CRP) observed only in female players (*Kostrzewa-Nowak et al., 2015*).

Team sports like football involve alternating high and low intensity efforts, blending anaerobic and aerobic exercise demands (*Mohr, Krustrup & Bangsbo, 2003*). The ability to perform repeated sprints with limited recovery, known as repeat sprint ability (RSA), is crucial for players (*Edge et al., 2005; Hill-Haas et al., 2007*). Female players typically exhibit lower performance in VO2 max and RSA tests compared to male players (*Sanders et al., 2017*). *Dent et al. (2015)* investigated sex differences in eukaryotic translation-initiation factor 4E binding protein 1 (4E-BP1) activity, metabolic markers, and RSA. 4E-BP1 is a key regulator of translation initiation, undergoing phosphorylation at the threonine 37/46 sites (Thr37/46) (*Ayuso et al., 2015*). The study found higher total 4E-BP1 at rest in female players than in male players (*Dent et al., 2015*). The ratio of phosphorylated 4E-BP1 Thr37/46 to total 4E-BP1 Thr37/46 was greater in male players at rest. Only males showed a decrease in this ratio post-exercise, suggesting sex differences in the metabolic mechanisms regulating muscle mass. Male players had faster sprint speeds, but this difference was offset when normalized by VO2 max. However, male players showed less decline in sprint performance, indicating better fatigue resistance. During and after exercise, physiological responses like heart rate and blood lactate increased similarly in both sexes, indicating a comparable reliance on anaerobic metabolism.

Elite football players exhibited similar post-match physiological responses across sexes, with levels of Interleukin 6 (IL-6) and tumor necrosis factor alpha (TNF-α) rising significantly post-match and normalizing within 24 h. CK and CRP peaked at 24 h, with CRP returning to baseline by 48 h. Male players had higher TNF-α levels immediately post-match (*Souglis et al., 2015*). Five days later, CK and uric acid levels had not fully recovered in all players, with male midfielders and forwards still showing elevated levels of protein carbonyls (*Souglis et al., 2018*). While both sexes experienced muscle damage and inflammation, female players had milder responses, potentially due to lower muscle load and estrogen's protective effects on cell membranes (*Datson et al., 2014; Gabbett & Mulvey, 2008*).

*McFadden et al. (2024)* found that IL-6 and CK levels rose with training load in all football players throughout the season, with males showing higher CK levels. Female players maintained stable estrogen, free testosterone, and total testosterone (tT) levels, while male estrogen and tT levels declined. Potentially due to estrogen's effect on growth hormone (GH) pulsatility (*Veldhuis, 1998*), female players had higher GH levels (*McFadden et al., 2024*). Insulin-like growth factor 1 levels increased in females and

decreased in males later in the season (*McFadden et al., 2024*). Total cortisol levels were consistently higher in female players. There was a trend towards iron (Fe) deficiency or anemia in female players, with male players maintaining higher ferritin levels. *Rodas et al. (2022)* observed significant metabolic differences between sexes, but sampling size and timing varied.

## Physical performance (*n* = 12)

The stretch-shortening cycle (SSC), which naturally pairs eccentric with concentric muscle movements to enhance power, is crucial for sports performance (*Cavagna, Saibene & Margaria, 1965*; *Norman & Komi, 1979*). Football players' ability in the SSC is commonly evaluated using squat jump (SJ) and counter-movement jump (CMJ) tests (*Markovic et al., 2004*). Female players exhibit lower explosive performance in these tests compared to males (*Cardoso de Araujo et al., 2018*; *McFarland et al., 2016*; *Mujika et al., 2009*; *Suchomel et al., 2015*; *Suchomel, Sole & Stone, 2016*), which may be attributed to lower testosterone levels and the resulting differences in muscle mass and strength (*Hubal et al., 2005*; *West & Phillips, 2010*). These disparities are also evident in sprinting ability (*Baumgart, Freiwald & Hoppe, 2018*; *Cardoso de Araujo et al., 2018*; *Devismes et al., 2019*; *McFarland et al., 2016*; *Mujika et al., 2009*). *Baumgart, Freiwald & Hoppe (2018)* found that the sprint mechanical properties of female Bundesliga players during the 30-m sprint were less efficient compared to male players and were akin to those of U14 and U15 male players.

*Sheppard & Young (2006)* identified four factors influencing change-of-direction (COD) performance in football: anthropometry, leg muscle quality, straight-sprint speed, and technique. These factors likely explain the consistent sex differences in COD abilities observed in the literature (*Condello et al., 2016*; *McFarland et al., 2016*; *Mujika et al., 2009*; *Nagano et al., 2016*). The first three factors have been discussed. Technical differences, such as insufficient trunk lean and a smaller femoral inclination angle in female players, which may result in a relatively stiff cutting movement (*Nagano et al., 2016*), also contribute to sex differences in COD abilities.

The interval shuttle run has become a key measure of a football player's endurance, vital for both assessment and training (*Krustrup et al., 2003*, *2005*). *Dolci et al. (2021)* found that female players exhibited superior movement economy in 10 m shuttle runs. The study's protocol included three sets of 5-min runs at 8.4 km/h, interspersed with recovery periods. However, the protocol's ecological validity for elite players is questionable, as the sustained steady low-speed running is not typical of match play. Football endurance is high-intensity intermittent running (*Bradley et al., 2009*; *Di Salvo et al., 2009*). The Yo-Yo intermittent recovery (Yo-Yo IR) test involves 20 m shuttle runs at progressively increasing speeds, with 10 s active rest intervals. The test continues until the participant is exhausted (*Krustrup et al., 2003*). *Mujika et al. (2009)* found that female players completed only half the distance of males in the Yo-Yo IR1 test. Moreover, three studies examined sex differences using the Interval Shuttle Run Test (ISRT) (*Baumgart, Hoppe & Freiwald, 2014*; *Cardoso de Araujo et al., 2019*; *Cardoso de Araujo et al., 2018*). During the ISRT, participants run and walk in 30 s and 15 s cycles on 20-m and 8-m tracks, respectively, with speed increments every 90 s until the pace cannot be maintained (*Lemmink et al., 2004*). Two studies also included an

incremental endurance test based on blood lactate levels to assess continuous endurance (*Baumgart, Hoppe & Freiwald, 2014*; *Cardoso de Araujo et al., 2018*). The contribution of anaerobic metabolism is greater during the ISRT compared to the incremental test. Sex differences are more pronounced in intermittent and non-linear running performance (*Baumgart, Hoppe & Freiwald, 2014*; *Cardoso de Araujo et al., 2018*). *Baumgart, Hoppe & Freiwald (2014)* suggested that intermittent shuttle run performance better explains sex differences in high-intensity running during matches than lactate thresholds.

## Technical performance (*n* = 14)

Female players are generally reported to achieve lower ball velocities and foot velocities than male players (*Barfield, Kirkendall & Yu, 2002*; *Iitake et al., 2022*; *Navandar, Kipp & Navarro, 2022*; *Sakamoto & Asai, 2013*), although there are individual exceptions (*Barfield, Kirkendall & Yu, 2002*; *Orloff et al., 2008*). Accurate kicks had lower peak knee velocities than unsuccessful kicks for both sexes (*Gheidi & Sadeghi, 2010*). *Iitake et al. (2022)* found that female players used a smaller knee extension moment with and relied more on hip flexion moment for the instep kick, whereas *Navandar, Kipp & Navarro (2022)* found no sex differences in hip and knee joint moments. The discrepancy might stem from variations in participants and run-up techniques between the studies. Despite no sex differences in plant leg position, female players exhibited greater trunk tilt, plant leg angle, and lateral and medial ground reaction forces (GRF), whereas male players demonstrated higher vertical GRF at ball contact (*Orloff et al., 2008*).

Female players head the ball less in matches than male players (*Jackson et al., 2023*; *Langdon et al., 2022*; *Nelson et al., 2020*; *Peek et al., 2024*; *Reynolds et al., 2017*; *Saunders et al., 2020*), with defenders leading in both sexes (*Langdon et al., 2022*; *Nelson et al., 2020*). However, header frequency is similar in training (*Reynolds et al., 2017*). Weaker head-neck strength in female players may lead to greater head impact kinematics (*Bretzin et al., 2016*). No acute cognitive impact from heading was observed for both sexes (*Putukian, Echemendia & Mackin, 2000*), but long-term effects require further investigation. Female players more often position their bodies optimally for headers (*Jackson et al., 2023*), but tend to close their eyes earlier than males before heading (*Peek et al., 2024*). Male players demonstrated better head control, used the forehead more, and relied on upper body strength more often (*Peek et al., 2024*). Additionally, females are more likely to perform headers from corners and goal kicks, whereas males more frequently engage in heading from free play, with the purpose of heading also differing by sex: females tend to use it for intercepting play, and males for passing the ball (*Peek et al., 2024*).

## Match performance (*n* = 5)

Female players are frequently reported to experience more stoppage events than males (*Pappalardo et al., 2021*; *Siegle & Lames, 2012*), particularly throw-ins (*Siegle & Lames, 2012*). This may result from male players' generally more accurate passing and longer kicking distances (*Bradley et al., 2014*; *Pappalardo et al., 2021*). However, female players take less time to restart play and have a lower incidence of prolonged stoppages than males (*Pappalardo et al., 2021*; *Siegle & Lames, 2012*). Furthermore, the number of fouls in female

matches is typically lower than in male matches (*Pappalardo et al., 2021*). A pattern seen in both male and female matches shows that leading teams attempt to shorten the match, while trailing teams hasten to resume play (*Siegle & Lames, 2012*).

*Bradley et al. (2014)* reported no sex differences in technical aspects like ball touches and possession in matches of the European Champions League, but found females covered less running distance than males. In contrast, *McFadden et al. (2020)* found no sex-based distance difference in American collegiate players. This discrepancy may stem from differences in competition level and time of publication, with potential improvements in female training. Both studies noted increased sex differences at higher running speeds, with female players engaging in fewer sprints (*Bradley et al., 2014*; *McFadden et al., 2020*). However, *Jastrzębski & Radzimiński (2017)* argued that default speed zones may undervalue female high-intensity running, showing in small-sided games that male players covered more distance but females ran a higher proportion of their total distance at high intensities above the velocity at lactate threshold.

## Psychological and behavioral performance (*n* = 7)

In terms of visual attention, *Jin et al. (2023)* found no sex differences among elite players during a multiple object tracking task. Regarding penalty shootout behavior patterns, *Avugos et al. (2022)* observed that female kickers had a stronger tendency to kick towards the goalkeepers' right. In a study evaluating the mental health of football players, *Bonet et al. (2024)* found that all values fell within the normal range across both sexes, although female players reported higher scores only on the somatic anxiety variable as measured by the Hamilton Anxiety Rating Scale (HARS) (*Lobo et al., 2002*). Moreover, female players' sport confidence is more influenced by factors such as mastery, demonstration of ability, preparation, self-presentation, social support, coach leadership, and vicarious experience than males (*Adegbesan, 2007*). Female players prefer coaches with good social skills, place more emphasis on social relationships, and show greater empathy towards their coaches. Conversely, male players place more importance on performance and experience greater satisfaction and motivation when they outperform other players (*López-Gajardo et al., 2021*). The presence of a competitor triggers a dominance instinct in males, driving them to aim for superior performance and focus more on match outcomes over the process (*De la Vega et al., 2022*). Although elite football inherently emphasizes competitiveness, match intensity may also escalate as a result, partially accounting for why male players exhibit more aggressive behaviors than their female counterparts (*Coulomb-Cabagno & Rascle, 2006*).

## Nutrition/Recovery/Sleep (*n* = 11)

*Sebastia-Rico et al. (2024)* observed that male players had higher fluid intake and sweating rates than females during summer. According to the Union of European Football Associations (UEFA) guidelines, players should consume 1.6–2.2 g/kg/day of protein (*Collins et al., 2021*), but *Kwon et al. (2023)* found that female players did not meet this recommendation, whereas male players did. Female players consumed less protein overall and had higher intakes of sugar, saturated fat, and vitamin C, but lower intakes of

potassium, sodium, iron, magnesium, vitamin D, and protein-rich foods (*Gomez-Hixson, Biagioni & Brown, 2020*). In terms of trace minerals, intake of molybdenum, zinc, and copper showed no sex differences (*Toro-Román et al., 2023b*, *2022*, *2024*). Female players had lower hemoglobin, erythrocytes, ferritin, and serum iron, but higher cadmium and lead levels in plasma, erythrocytes, and platelets (*Toro-Román et al., 2023c*). Male players had higher plasma and urine concentrations of molybdenum and zinc, whereas females had higher erythrocyte zinc levels (*Toro-Román et al., 2022*, *2024*). Additionally, males excreted more copper in their urine and had higher aluminum intake and plasma levels compared to females (*Robles-Gil et al., 2023*; *Toro-Román et al., 2023b*).

*Ros et al. (2013)* assessed players' recovery using the Yo-Yo IR2 test (*Bradley et al., 2011*) as a fatiguing protocol and found no sex differences in post-exercise jump performance trends. Both sexes showed a decrease in single-leg jumps performance and an improvement in square hop performance. *Koikawa et al. (2016)* evaluated players' sleep quality and sleepiness using the Pittsburgh Sleep Quality Index (*Buysse et al., 1989*; *Doi et al., 2000*) and the Epworth Sleepiness Scale (*Johns, 1991*; *Takegami et al., 2009*), finding that female players had poorer sleep quality and higher sleepiness than males. Conversely, *Biggins et al. (2022)* found no sex differences in sleep variables using actigraphy and the Karolinska Sleepiness Scale (*Åkerstedt, Hallvig & Kecklund, 2017*), though female players reported more pre-sleep anxiety. Pre-match sleep quality declined for all players.

## DISCUSSION

This scoping review aimed to synthesize evidence on sex differences among elite football players and identify research gaps to guide future studies. Eighty studies were included, covering 2,226 female and 2,670 male players, and 99 female and 135 male matches. Previous research has highlighted a significant sex gap in football-related studies (*Okholm Kryger et al., 2021*), and the differing sample sizes in this review underscore the underrepresentation of female players, further emphasizing the need for more focused research on female players. Moreover, among the included studies, there was either confusion between the terms "sex" and "gender", a lack of reporting on female players' menstrual cycles and contraceptive use, or both. This highlights the pressing need to strengthen methodological and terminological standards in future research. Although the included studies in this scoping review are categorized into eight themes, there are relationships and connections between each theme. This is evident in the overlapping studies identified in the review across different themes, which means that conducting research in one area has the potential to impact another in practice and research (*Whitehead et al., 2021*). Additionally, there is relatively little research in some more specific themes, which makes it difficult to draw unequivocal conclusions.

According to anthropometric research, female football players typically exhibit a higher percentage of body fat and lower lean body mass compared to their male counterparts (*Ahmad & Abu Al Haija, 2024*; *Mascherini et al., 2017*; *Petri et al., 2024a*, *2024b*; *Schons et al., 2022*; *Tornero-Aguilera, Villegas-Mora & Clemente-Suarez, 2022*; *Toro-Román et al., 2023a*). Additionally, they possess distinct skeletal characteristics, including smaller bone size and lower bone density (*Baker et al., 2020*; *Hedt et al., 2022*). However, the results of

%BF can be influenced by varying estimation and measurement methods. *Tornero-Aguilera, Villegas-Mora & Clemente-Suarez (2022)* reported that bioelectrical impedance analysis underestimates the %BF, followed by skinfold thickness measurements (SKF), with dual energy X-ray absorptiometry (DEXA) being more objective and accurate. Nevertheless, the high cost of DEXA equipment limits its widespread accessibility. UEFA has recommended using absolute SKF to evaluate changes in body composition, as opposed to using equations to calculate %BF (*Collins et al., 2021*). Only two studies have examined skeletal characteristics (*Baker et al., 2020*; *Hedt et al., 2022*), and future research should focus on this area by incorporating larger sample sizes to allow for more direct and reliable assessments of raw anthropometric measures. This will establish a baseline for understanding the physical characteristics of elite football players of different sexes (*Santos et al., 2014*). Additionally, since some anthropometric parameters change throughout the season, particularly due to varying training loads and different phases of the menstrual cycle (*Campa et al., 2021*; *Clemente, Ramirez-Campillo & Sarmento, 2021*), subsequent research should monitor and evaluate these changes over the entire season to enable longitudinal comparisons of sex differences.

Muscle and joint movement research highlighted that elite female football players encounter a dual aspect of difference (*Brophy et al., 2009*; *Burfeind, Hong & Stavrianeas, 2012*; *Hart et al., 2007*; *Hewett et al., 2006*; *Ichinose et al., 1998*; *Kanehisa et al., 2003*; *Makaraci et al., 2024*; *Yilmaz et al., 2023*; *Zebis et al., 2011*). On one hand, they exhibit a gap in absolute muscle strength compared to male players, which may be due to differences in muscle cross-sectional area. On the other hand, bilateral muscle asymmetries, as well as core strength deficits, can put them at a higher risk of injury. Future research could focus on developing specific training programs tailored for female players to reduce asymmetries and thereby lower the risk of injury. Alternatively, it can explore whether trunk stability in female players is associated with their distinct pelvic anatomy.

Three studies investigated the cardiovascular adaptive changes in players of different sexes (*Mascherini, Petri & Galanti, 2018*; *Sansonio de Morais et al., 2018*; *Steding et al., 2010*). Their results align with the findings reported in systematic reviews involving a broader exercise population (*Bassareo & Crisafulli, 2020*). Sex may influence physiological cardiovascular regulation and adaptation to exercise, with genetics, endocrine factors, and body composition characteristics potentially being key determinants of these differences (*Bassareo & Crisafulli, 2020*). When cardiovascular variables are normalized for body surface area, many sex differences are attenuated or even abolished, suggesting that future research should employ relative values rather than absolute values. The remaining ten physiological studies examined sex differences in the acute and long-term responses to anaerobic or aerobic exercise, as well as to competitive matches (*Chamera et al., 2014*, *2015*; *Dent et al., 2015*; *Kostrzewa-Nowak et al., 2015*; *Magal et al., 2020*; *McFadden et al., 2024*; *Rodas et al., 2022*; *Sanders et al., 2017*; *Souglis et al., 2018*; *Souglis et al., 2015*). Consistent with previous research (*Ansdell et al., 2020*), integrative metabolic thresholds during exercise are influenced by phenotypical sex differences across many physiological systems. One key contributing factor to the sex differences observed in metabolic responses to exercise may be the higher percentage of essential body fat in females

compared to males (*Lewis, Kamon & Hodgson, 1986*). However, none of those ten physiological studies controlled for the use of oral contraceptives among female players. Additionally, the menstrual cycle was not controlled for. These two points were overlooked in the majority of the studies included in this review. The absence of controls may skew the results, given that hormonal fluctuations are known to affect various physiological responses (*Datson et al., 2014*). To overcome these limitations, future research should standardize the timing of sample collection relative to the menstrual cycle and control for the use of oral contraceptives. This will enable a more accurate comparison of physiological responses between male and female football players and facilitate a deeper investigation into the regulatory mechanisms governing the acute responses in football players. Similarly, *Sims & Heather (2018)* have noted that sports science research focusing on female players remains limited, in part due to methodological shortcomings in study design that fail to account for reproductive status (*e.g.*, menstrual cycle phase, hormonal contraception use). By explicitly characterizing hormonal fluctuations across distinct physiological phases and their systemic effects, methodological inconsistencies can be reduced. This refinement in experimental rigor will facilitate the evaluation of sex-based differences in exercise-related physiological responses.

Perhaps due to biological differences (*Miller et al., 1993*; *Nuzzo, 2022*), particularly in neuromuscular strength, female players generally score lower than males in assessments of physical performance (*Baumgart, Freiwald & Hoppe, 2018*; *Baumgart, Hoppe & Freiwald, 2014*; *Cardoso de Araujo et al., 2019*, *2018*; *Condello et al., 2016*; *Devismes et al., 2019*; *Dolci et al., 2021*; *McFarland et al., 2016*; *Mujika et al., 2009*; *Nagano et al., 2016*; *Suchomel et al., 2015*; *Suchomel, Sole & Stone, 2016*). However, these sex differences may precisely reflect the unique physical demands of female football, suggesting that future research should develop a physical evaluation system aligned with these demands. Furthermore, given these biological differences, future research should propose targeted training protocols based on the sex-specific characteristics of female players. Additionally, more studies could focus on validating the effectiveness of long-term intervention measures for enhancing physical performance, especially considering that existing research is primarily short-term and cross-sectional. While improving match performance is the common ultimate goal for both sexes, the optimal pathway to achieve this goal appears to vary by sex (*Cardoso de Araujo et al., 2018*).

Studies on technical performance differences mainly concentrate on kicking and heading. All studies involving kicking techniques have conducted biomechanical analyses of the instep kick to compare kinematic and kinetic sex differences (*Barfield, Kirkendall & Yu, 2002*; *Gheidi & Sadeghi, 2010*; *Iitake et al., 2022*; *Navandar, Kipp & Navarro, 2022*; *Orloff et al., 2008*; *Sakamoto & Asai, 2013*). However, current research primarily focuses on the instep kicking technique. Future research could explore the differences exhibited by male and female soccer players when kicking with various parts of the foot. In contrast to extracting kicking techniques for biomechanical analysis in laboratory settings to investigate technical performance, research on heading techniques tends to focus more on the frequency and impact of headers during actual matches and training (*Bretzin et al., 2016*; *Jackson et al., 2023*; *Langdon et al., 2022*; *Nelson et al., 2020*; *Peek et al., 2024*;

*Putukian, Echemendia & Mackin, 2000*; *Reynolds et al., 2017*; *Saunders et al., 2020*). Only one study has focused on the acute effects of heading on cognitive function in male and female players, reporting no significant acute changes in either group (*Putukian, Echemendia & Mackin, 2000*). This result is consistent with other review studies (*Kontos et al., 2016*; *McCunn et al., 2021*; *Shen et al., 2024*). However, the observed negative impacts are limited to specific outcome measures (*McCunn et al., 2021*), underscoring the necessity for broader and more in-depth investigations in future studies.

Research on sex differences in match performance remains scarce, with two studies focusing on stoppages and resumptions of play (*Pappalardo et al., 2021*; *Siegle & Lames, 2012*), and three examining running characteristics during matches (*Bradley et al., 2014*; *Jastrzębski & Radzimiński, 2017*; *McFadden et al., 2020*). Consistent with previous research findings (*Bradley & Vescovi, 2015*), the current velocity thresholds used for analyzing male matches may distort the observation of high-intensity movement demands in female matches, as these thresholds do not accurately reflect the capabilities of elite female players (*Bradley et al., 2014*; *Jastrzębski & Radzimiński, 2017*; *McFadden et al., 2020*). Future studies could establish sex-specific benchmarks through large-scale normative data collection, or validate individualized velocity thresholds based on physiological measures to more accurately assess the performance of female players.

The current psychological and behavioral studies have limited sample sizes and scope (*Adegbesan, 2007*; *Avugos et al., 2022*; *Bonet et al., 2024*; *Coulomb-Cabagno & Rascle, 2006*; *De la Vega et al., 2022*; *Jin et al., 2023*; *López-Gajardo et al., 2021*), especially given the high heterogeneity in this field. Therefore, further research is necessary to refine and expand these findings. These studies do offer valuable insights into the psychological and behavioral traits of football players across different sexes. For example, when working with female players, coaches and practitioners might focus on group-based tasks, whereas they could emphasize the competitive aspects of tasks when engaging with male players (*López-Gajardo et al., 2021*).

Nutritional research has shown that female players not only consume less protein than their male counterparts but also fail to meet the recommended dietary guidelines (*Gomez-Hixson, Biagioni & Brown, 2020*; *Kwon et al., 2023*). Future research could focus on investigating the factors that influence dietary intake through larger sample sizes and evaluating the efficacy of nutritional education interventions. Additionally, it could determine whether long-term inadequate dietary intake translates into actual deficiencies and subsequently impairs performance and long-term health in players (*Gomez-Hixson, Biagioni & Brown, 2020*). Only one study has investigated sex differences in the recovery from fatigue in elite football players (*Ros et al., 2013*), highlighting a gap in this research area. Future research can focus on the sex-specific mechanisms of fatigue recovery in football players and test specific strategies optimized for the fatigue recovery of different sexes. The lack of existing research is also evident in studies on sex differences in sleep quality (*Biggins et al., 2022*; *Koikawa et al., 2016*). Future research should replicate and validate these findings using larger and more diverse samples, and investigate the mechanisms underlying sex differences in sleep.

In summary, female football players generally exhibit a higher percentage of body fat and lower lean body mass compared to male players, and have different skeletal characteristics, such as smaller bone size and lower bone density (*Ahmad & Abu Al Haija, 2024*; *Baker et al., 2020*; *Hedt et al., 2022*; *Mascherini et al., 2017*; *Petri et al., 2024a*, *2024b*; *Schons et al., 2022*; *Tornero-Aguilera, Villegas-Mora & Clemente-Suarez, 2022*; *Toro-Román et al., 2023a*). Regarding muscle and joint mechanics, female players often demonstrate poorer core control, imbalances in hip muscle strength, and bilateral asymmetry in muscle power compared to their male counterparts, factors which may increase their risk of ACL and other injuries (*Brophy et al., 2009*; *Hart et al., 2007*; *Hewett et al., 2006*; *Yilmaz et al., 2023*). Although female players have lower aerobic and mixed anaerobic capacities than males, both sexes experience similar muscle damage and inflammation after official matches, with female players tend to have milder responses (*Chamera et al., 2014*, *2015*; *Dent et al., 2015*; *Kostrzewa-Nowak et al., 2015*; *Magal et al., 2020*; *Rodas et al., 2022*; *Sanders et al., 2017*; *Souglis et al., 2018*, *2015*). However, significant sex differences exist in explosive power, sprinting ability, COD ability, and intermittent endurance (*Baumgart, Freiwald & Hoppe, 2018*; *Baumgart, Hoppe & Freiwald, 2014*; *Cardoso de Araujo et al., 2019*, *2018*; *Condello et al., 2016*; *Devismes et al., 2019*; *Dolci et al., 2021*; *McFarland et al., 2016*; *Mujika et al., 2009*; *Nagano et al., 2016*; *Suchomel et al., 2015*; *Suchomel, Sole & Stone, 2016*). In terms of kicking technique, female players generally achieve lower ball velocities and foot velocities (*Barfield, Kirkendall & Yu, 2002*; *Iitake et al., 2022*; *Navandar, Kipp & Navarro, 2022*; *Sakamoto & Asai, 2013*). Female players use heading less frequently than male players, and the purpose of heading also differs between the sexes (*Jackson et al., 2023*; *Langdon et al., 2022*; *Nelson et al., 2020*; *Peek et al., 2024*; *Reynolds et al., 2017*; *Saunders et al., 2020*). Additionally, in matches, female players experience more frequent stoppages (*Pappalardo et al., 2021*; *Siegle & Lames, 2012*), cover less total distance, and perform fewer sprints compared to males (*Bradley et al., 2014*; *Jastrzębski & Radzimiński, 2017*; *McFadden et al., 2020*). Psychologically and behaviorally, female players tend to place greater importance on team relationships and show more empathy towards their coaches. In contrast, male players tend to focus more on match outcomes rather than the process itself, and exhibit more aggressive behaviors during matches (*López-Gajardo et al., 2021*). Regarding nutrient intake, female players often consume more sugar and saturated fat but fall short of recommended protein intake (*Gomez-Hixson, Biagioni & Brown, 2020*; *Kwon et al., 2023*). In terms of sleep quality, female players may experience relatively poorer sleep quality and report more pre-sleep anxiety (*Biggins et al., 2022*; *Koikawa et al., 2016*).

## Limitations

This review has certain limitations that must be acknowledged. Firstly, while this review adhered to PRISMA-ScR (*Tricco et al., 2018*) guidelines, the included studies were limited to four databases and the established inclusion and exclusion criteria. It is important to note that relevant research may exist beyond these databases, in other languages, or in formats other than scientific articles. Secondly, this review focuses on sex differences within the binary classification of elite football players. The included studies also adhered

to this classification, despite variability in the terminology used for sex. The decision to use sex-related terms in this review stems from the focus on biological factors in the majority of studies and to maintain terminological consistency. However, in some psychological studies, the use of gender-related terms might have been more suitable. Finally, the included studies were manually categorized into thematic groups by the research team. While this process was independently performed by two researchers and final consensus was achieved through discussion, potential subjective bias in the classification cannot be entirely ruled out.

## CONCLUSIONS

This study is a scoping review of sex differences in elite football players. A total of 80 studies were included, demonstrating the differences between female and male players across various themes. Overall, perhaps due to a combination of innate biological factors and acquired dietary habits, female players, who typically have a higher body fat percentage and lower lean body mass, exhibit performance gaps compared to male players in terms of speed, strength, and endurance. These differences further affect their technical skills and match performance. However, while these absolute differences exist, they often diminish or even disappear when data are standardized against certain anthropometric or physiological metrics. This underscores the importance of developing individualized analytical methods and evaluation criteria tailored to female players. Furthermore, this review highlighted limitations in current research, including the confusion of sex and gender terminology and the failure to account for reproductive status. Future studies should carefully define sex-specific inclusion/exclusion criteria and select appropriate sex and gender terminology to minimize bias and enhance study quality.

### Funding
The authors received no funding for this work.

### Competing Interests
The authors declare that they have no competing interests.

### Author Contributions
- Wangyang Xu conceived and designed the experiments, performed the experiments, analyzed the data, prepared figures and/or tables, authored or reviewed drafts of the article, and approved the final draft.
- Diyan Zhang conceived and designed the experiments, performed the experiments, analyzed the data, authored or reviewed drafts of the article, and approved the final draft.
- Xinbi Zhang conceived and designed the experiments, performed the experiments, analyzed the data, authored or reviewed drafts of the article, and approved the final draft.
## Data Availability

This is a literature review.

## Supplemental Information

Supplemental information for this article can be found online at http://dx.doi.org/10.7717/peerj.19976#supplemental-information.

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
