# Peer review of "Sex differences in association football: a scoping review"

_PeerJ, doi:10.7717/peerj.19976_

## Round 0.1 · original submission · Major Revisions

Overall, the concept of your review and conduct is well received. There are areas for improvement in the discussion, future directions, and review limitations. Please see the reviewer's comments to assist with making amendments to these sections.

Reviewer 1 ·

Basic reporting

All relevant observations are explicitly highlighted in the "additional comments section."

Experimental design

All relevant observations are explicitly highlighted in the "additional comments section."

Validity of the findings

All relevant observations are explicitly highlighted in the "additional comments section."

Additional comments

Overall impressions
The authors synthesized the available body of evidence referring to the sex differences in various outcomes among elite football players. Indeed, it is essential to highlight that the addressed topic is quite relevant and useful, including the overall scientific community, practitioners, and wider readership of the journal. Moreover, examining the sex differences, particularly among football players, is a potentially highly cited study, consequently positively affecting the impact factor of the journal in the prestigious databases. On the other hand, numerous aspects of the study should be significantly modified, especially the clarity and interpretation of the most relevant findings in the results and discussion section. Still, the expertise of the authors is most likely appropriate with respect to the successful modifications of the manuscript in line with all remarks available below.
Specific comments
• Lines 20-21: The overall methodology of the abstract section should be substantially improved. More specifically, several relevant aspects referring to the methods currently are not highlighted in the text. Therefore, please provide the necessary correction with respect to the mentioned crucial segment of the abstract of the present study.
• Lines 22-28: The revealed future directions and essential findings of this investigation should be explicitly and comprehensively emphasized in the results section. Please take into account this suggestion and modify the manuscript accordingly.
• Lines 29-35: The conclusion of the abstract section is considered indeed extensive. Please emphasize the fundamental aspects of the obtained results as well as the most relevant recommendations for future studies, providing between one and two statements.
• Collectively, the names of all addressed outcomes need to be more exactly and clearly highlighted across certain aspects of the study, including the introduction, methods, and results section. Please thoroughly check the overall manuscript and ensure the required changes.
• Lines 80-83: The currently available body of evidence referring to the examined topic should be more comprehensively analyzed in the last paragraph of the introduction. In other words, it is highly suggested to briefly underline the fundamental aspects with respect to the findings of the existing literature. Moreover, regarding the authenticity, the compatibility in the context of the assessed outcomes between the current study and the available body of knowledge should also be emphasized in the text. Please provide the suggested modifications.
• Lines 84-86: From my viewpoint, the statement pertaining to the definition of a scoping review is quite redundant. On the other hand, if the authors consider that the overall manuscript and the readership of the journal would have certain benefits concerning the definition of the design of the study, this aspect should be highlighted in the methods section.
• Lines 89-93: The practical aspect of the rationale of the study should be available in the text before statements referring to the goals. Please provide the structural corrections in the last paragraph of the introduction of the present investigation.
• Lines 95-99: The name of the first subheading of the methods section should be "Study design and protocol registration," including the statements with respect to the adherence to the PRISMA-ScR guidelines and registration via OSF.
• A more accurate and extensive analysis of the crucial aspects of the current scoping review, such as examined outcomes, is highly recommended in the text. Please modify the PI/ECOS and data items sections according to the highlighted observation.
• Line 117: The table regarding the overall search strategy should be given as online supplementary material. Please provide the necessary changes.
• Data synthesis is commonly an indispensable aspect of the methods of the scoping reviews. Therefore, please check the reputable body of scientific literature with respect to the mentioned design of the study and emphasize several statements pertaining to the synthesis of the findings.
• The innovative and current PRISMA 2020 flow diagram should be available in the results section. Please modify Figure 1 in line with the emphasized recommendation.
• Taking into account the text concerning the "methods" of Table 1S, the name and the overall definition of this aspect should be substantially modified. Specifically, are the methods referring to the applied measuring tools and examined outcomes of the included studies? Please clarify that.
• Line 201: The evaluation of the "blinding of assessors" domain is quite controversial, particularly considering that all involved studies were cross-sectional. Please provide a comprehensive explanation with respect to the emphasized aspect of the methodological quality of the present investigation.
• All the extensively described findings of the existing literature in the discussion should be merged with the results section. In other words, it is suggested for the authors to provide an independent subheading with the name "Results and Discussion." Most importantly, the highlighted approach is indeed common in the prestigious body of scientific literature. Finally, please consider improving the design and the overall understanding of the mentioned segments of the study.
• The fact is that the sex differences among football players were determined regarding the numerous outcomes, depending on the available subheadings. Hence, with regard to the improvements in the clarity of the obtained results of the study, it is suggested that the authors provide an additional separate subheading summarizing the most relevant findings with respect to all the assessed parameters. Please consider this recommendation.
• Currently, the overall results of this scoping review are not compared with the existing body of knowledge in the discussion section. Please justify yourself.
• Since the crucial goal of the scoping review is to identify gaps in the available literature and consequently highlight valuable future directions, it would be quite useful to extend and deepen the overall recommendations for future studies, including all eight relevant outcomes.
• From my viewpoint, the limitations section is fairly scarce. Therefore, please provide several additional restrictions, particularly with respect to the methodology of the present investigation.

·

Basic reporting

808 / 5.000
The level of English is one that did not create problems for me in analyzing the article. The vast majority of the references were relevant to the idea presented, and the study may be of interest, even for interdisciplinary fields. The introduction is ok, but it suddenly moves from one concept to another. I recommend the authors a logical, continuous transition between the different concepts addressed. (Example: the authors discuss the different aspects of different studies, aspects related to differences in sports performance or different criteria of analysis, after which the authors suddenly move on to the confusion or misuse of the terms "sex" and "gender", without making a direct connection with the previous ideas, and then, again, return to differences in performance between women and men in performance football).

Experimental design

The content of the article falls within the aims and scope of the journal, and the authors' approach complies with the technical standard. The study presents clear benchmarks for replication, if anyone wishes to do so. I would recommend the authors to reorganize the paragraphs in terms of the flow of ideas, the direct links between them. Perhaps a rethink of the presentation of the Introduction section.

Validity of the findings

The conclusions section is far too simple and repeats ideas from the introduction. I would like the authors to highlight this section more clearly, with some scientific details to emphasize what is the originality of their ideas, what they brought new to the field, through this literature review.
Also, L488-489 does not represent a conclusion. It has no place in this section. Please remove it.

Additional comments

The idea is one to be appreciated, although the essence is somewhat known. It is clear that there are differences, normal ones, between different sexes, but it is appreciated that the authors' analysis draws attention to certain distinct aspects that can represent the basis for future research.

---

## Round 0.2 · Minor Revisions

Thank you for your efforts in addressing the reviewer feedback and comments. Before your work can be considered for publication, there are minor amendments required. Please address the comments of the reviewers regarding reporting, language use, and clarity.

Reviewer 1 ·

Basic reporting

Each relevant remark concerning the emphasized aspect of the study is available in the overall impression section.

Experimental design

Each relevant remark concerning the emphasized aspect of the study is available in the overall impression section.

Validity of the findings

Each relevant remark concerning the emphasized aspect of the study is available in the overall impression section.

Additional comments

Overall impressions regarding the revision of the manuscript
Thank you for your comprehensive feedback referring to the current investigation, including the high-quality response to all the remarks and the modifications of the relevant aspects of the manuscript. Indeed, it is crucial to emphasize that the authors appropriately corrected most of the controversies with respect to the analyzed study that are available in the review report. However, several segments of the abstract and discussion section still require substantial changes. More specifically, with regard to the abstract of the study, the most relevant findings referring to all the assessed outcomes/themes should be explicitly highlighted in the text (lines 31-35). Moreover, although the recommendations for future investigations in the conclusion are considered satisfactory, the fundamental aspects of the obtained results are not emphasized in the manuscript (lines 42-44). Most importantly, the overall context of the "general discussion" section is indeed questionable. In other words, the aspects concerning the comparison with the existing body of literature and future directions should be highlighted in the "results and discussion" of the present study. In addition, the subheading pertaining to the summary of the principal findings should encompass several statements (approximately 10), indicating the essence of the present scientific work. Overall, taking into account the effort and expertise of the authors, the previously mentioned controversies will most likely be successfully corrected, potentially fulfilling the high criteria of the journal and consequently providing acceptance as the final decision.
Best regards!

·

Basic reporting

I believe that the authors took into account the recommendations, paid attention to a more obvious logic and flow, compared to the first form of the manuscript. I have no other observations, given that it is a scoping review, correctly approached in terms of methodology and data presentation.
However, I also consider these recent studies relevant, on the analysis of various functional, morphological and motor parameters in football. I recommend the authors to analyze these references:
- Alexe DI, Čaušević D, Čović N, Rani B, Tohănean DI, Abazović E, Setiawan E, Alexe CI. The Relationship between Functional Movement Quality and Speed, Agility, and Jump Performance in Elite Female Youth Football Players. Sports. 2024; 12(8):214. https://doi.org/10.3390/sports12080214,
- Čaušević D, Rani B, Gasibat Q, Čović N, Alexe CI, Pavel SI, Burchel LO, Alexe DI. Maturity-Related Variations in Morphology, Body Composition, and Somatotype Features among Young Male Football Players. Children. 2023; 10(4):721, https://doi.org/10.3390/children10040721,

Experimental design

I have no other recommendations or observations. I believe the study design is one that meets the standards.
I would like the authors to specify in their manuscript whether they considered as an inclusion or exclusion criterion the articles that presented studies and analyses on male and female persons with disabilities? Were these studies included? Were they excluded? Regardless of inclusion or exclusion, this aspect should be specified, CLEARLY, in the manuscript.

Validity of the findings

The objectives proposed for this stage were achieved, the authors presenting synthetically what they set out to do by orienting the discussion towards the central idea: the differences existing and determined by other studies and research regarding female and male subjects in elite football. BUT:

1. In relation to the criterion "healthy as a subject" and "disability", this conclusion ("This study provides the first comprehensive overview of sex differences in football") should be reformulated, to refer to sex differences but to the clear category of subjects, taking into account the classification criteria discussed previously. Maybe the authors include this disability criterion as a limit?!! they will decide, but it must be addressed, to be eliminated or included in the general discussion.

2. I stick to the idea from the previous report. And I maintain it strongly. The conclusions must clearly summarize the idea from the title and purpose. If the title and the proposed purpose are to highlight the differences, I ask the authors, through the conclusions, to bring to the readers what are the biggest differences discovered by this study, between the subjects, male and female, practitioners of performance football.
Why? if a reader does not want to read the entire article, but the title and purpose attract his attention, he will want to read the conclusions ...... to see where the authors have arrived and what is the essence of the study (what are the clear, net, visible, essential differences) regarding these gender differences in performance football players.
I ask the authors to summarize a clear conclusion on this aspect.

Additional comments

what are these sex differences?? ok, it is known that they exist, there are studies that present them, but this review must also end with an opinion of the authors themselves, a conclusion of their own, generated by the analysis of so many studies.

---

## Round 0.3 · Minor Revisions

Please address reviewer comments about the length of the abstract and the specific statements that are suggested by the reviewer. You should also consider modifying the conclusion to summarise th key points of your article with appropriate direction for application in practice and future research.

Reviewer 1 ·

Basic reporting

All observations with respect to the present study are available in the "Additional comments" section.

Experimental design

All observations with respect to the present study are available in the "Additional comments" section.

Validity of the findings

All observations with respect to the present study are available in the "Additional comments" section.

Additional comments

Overall impressions regarding the revision of the manuscript
Thank you for your extensive and dedicated work in the context of all previously emphasized remarks referring to the abstract, results, and discussion sections. Namely, the available text unambiguously indicates that all the aspects of the results and discussion are modified in line with the relevant requirements. Therefore, the overall level of the mentioned segments of the content of the present study is considered satisfactory. However, the abstract section still needs certain corrections. More specifically, as already highlighted, the essential findings with respect to the evaluated outcomes/themes should be explicitly and more exactly available in the text. In addition, several statements concerning the characteristics of the included studies and quality assessment are redundant. Moreover, the overall conclusion is quite extensive. It is suggested that the authors provide only the most relevant aspects of the conclusion, including the findings pertaining to the analyzed outcomes, practical implications, and future directions (2 or 3 statements). Finally, the text in the abstract section should not exceed 350 words.

---

## Round 0.4 · accepted · Accept

Thank you for your efforts in addressing reviewer feedback. I am pleased to advise that your manuscript is now suitable for publication.

Reviewer 1 ·

Basic reporting

All existing aspects of the relevant section are considered appropriate.

Experimental design

All existing aspects of the relevant section are considered appropriate.

Validity of the findings

All existing aspects of the relevant section are considered appropriate.

Additional comments

There are no additional comments with respect to the present study.

·

Basic reporting

I believe that the standards are met. The introduction sufficiently argues the subject and the proposed purpose. The references are, for the most part, relevant, although the authors were also offered more recent and appropriate ones. Overall, the theoretical basis is supported by the references.

Experimental design

I have no further comments to recommend.

Validity of the findings

I have no further comments to recommend.

Additional comments

I have no further comments to recommend.